# Real-World Experience of the One-Year Efficacy of Rifaximin Add-On to Lactulose Is Superior to Lactulose Alone in Patients with Cirrhosis Complicated with Recurrent Hepatic Encephalopathy in Taiwan

**DOI:** 10.3390/jpm11060478

**Published:** 2021-05-27

**Authors:** Ching Chang, Chien-Hao Huang, Hsiao-Jung Tseng, Fang-Chen Yang, Rong-Nan Chien

**Affiliations:** 1Division of Hepatology, Department of Gastroenterology and Hepatology, Chang-Gung Memorial Hospital, Linkou Medical Center, Taoyuan 33305, Taiwan; o78714@gmail.com (C.C.); huangchianhou@gmail.com (C.-H.H.); whiteeva907@gmail.com (F.-C.Y.); 2College of Medicine, Chang-Gung University, Taoyuan 33305, Taiwan; 3Biostatistics Unit, Clinical Trial Center, Chang Gung Memorial Hospital, Linkou Medical Center, Taoyuan 33305, Taiwan; allebjht@gmail.com

**Keywords:** rifaximin, lactulose, cirrhosis, hepatic encephalopathy, Taiwan

## Abstract

Background: Hepatic encephalopathy (HE), a neuropsychiatric complication of decompensated cirrhosis, is associated with high mortality and high risk of recurrence. Rifaximin add-on to lactulose for 3 to 6 months is recommended for the prevention of recurrent episodes of HE after the second episode. However, whether the combination for more than 6 months is superior to lactulose alone in the maintenance of HE remission is less evident. Therefore, the aim of this study is to evaluate the one-year efficacy of rifaximin add-on to lactulose for the maintenance of HE remission in Taiwan. Methods: We conducted a real-world single-center retrospective cohort study to compare the long-term efficacy of rifaximin add-on to lactulose (group R + L) versus lactulose alone (group L, control group). Furthermore, the treatment efficacy before and after rifaximin add-on to lactulose was also analyzed. The primary endpoint of our study was time to first HE recurrence (Conn score ≥ 2). All patients were followed up every three months until death, and censored at one year if still alive. Results and Conclusions: 12 patients were enrolled in group R + L. Another 31 patients were stratified into group L. Sex, comorbidity, ammonia level, and ascites grade were matched while age, HE grade, and model for end-stage liver disease (MELD) score were adjusted in the multivariable logistic regression model. Compared with group L, significant improvement in the maintenance of HE remission and decreased episodes and days of HE-related hospitalizations were demonstrated in group R + L. The serum ammonia levels were significantly lower at the 3rd and 6th month in group 1. Concerning changes before and after rifaximin add-on in group R + L, mini-mental status examination (MMSE), episodes of hospitalization, and variceal bleeding also improved at 6 and 12 months. Days of hospitalization, serum ammonia levels also improved at 6th month. Except for concern over price, no patients discontinued rifaximin due to adverse events or complications. The above results provide evidence for the one-year use of rifaximin add-on to lactulose in reducing HE recurrence and HE-related hospitalization for patients with decompensated cirrhosis.

## 1. Introduction

Hepatic encephalopathy (HE), a neuropsychiatric complication of decompensated cirrhosis, clinically manifests from minimal cognitive dysfunction to lethargy and, most seriously, coma [1]. HE is associated with high mortality and high risk of recurrence [1,2] and has a major impact on health-related quality of life in patients with liver cirrhosis [3].

There are various treatment choices for HE, including nonabsorbable disaccharides, antibiotics, and other potential therapies such as probiotics, branched-chain amino acids, and glutaminase inhibitors employing various mechanisms [1].

Lactulose, a nonabsorbable disaccharide, decreases the absorption of nitrogen-containing substances from the gastrointestinal tract via cathartic effects and reduces ammonia burden through converting ammonia to nonabsorbable ammonium by changing colonic pH [4]. The effect of lactulose for preventing overt HE is well reported [5], but adverse effects like severe diarrhea, bloating, flatulence, nausea, and vomiting may lower medication compliance [6].

Rifaximin, a poorly absorbed antibiotic, was approved in 2010 by the United States food drug administration (US FDA) for the treatment of overt hepatic encephalopathy via decreasing ammonia-generating enteric bacteria [7,8]. Its effects of reducing ammonia levels, preventing overt HE recurrence, as well as lowering HE-related hospitalization are documented in previous studies [9,10,11]. According to the current practice guideline by American association for the study of liver disease (AASLD)/European association for the study of liver (EASL), rifaximin add-on to lactulose over 3 to 6 months is recommended for the prevention of recurrent episodes of HE after the second episode [1]. Furthermore, rifaximin is reported to be safe for long-term use without obvious side effects due to characteristic of minimal absorption and fewer systemic effects [9,11]. 

However, the superiority of combination rifaximin and lactulose for more than six month over lactulose monotherapy alone in the maintenance of HE remission is less evident [12,13,14]. A few studies have investigated the prolonged effects of HE remission from a combination use of rifaximin with lactulose [13,15]. One study compared rifaximin with a placebo group rather than a lactulose group [16]. In addition, the use of rifaximin for the treatment of HE is less common in Taiwan because lactulose is supported by national health insurance administration while rifaximin is not.

Therefore, the aim of this study is to evaluate whether the one-year efficacy of rifaximin add-on to lactulose therapy is superior to lactulose alone for the maintenance of remission from overt hepatic encephalopathy in cirrhotic patients in Taiwan by using real-world data.

## 2. Materials and Methods

### 2.1. Study Design

We conducted a single-center retrospective cohort study to compare the long-term efficacy of rifaximin add-on to lactulose versus lactulose alone. In addition, the treatment efficacy regarding HE before and after rifaximin add-on to lactulose was also analyzed. This study was approved by the Institutional Review Board of Linkou Chang Gung Memorial Hospital, and informed consent was obtained from all patients in our study (201701810B0).

### 2.2. Patient Selection

From January 2015 to December 2019, consecutive patients age 18 years or older and diagnosed with liver cirrhosis complicated by HE (at least two episodes during the previous 6 months) were included. Patients with other diagnosed neurological or psychiatric comorbidities (like Alzheimer’s, Parkinson’s, stroke with neurological deficit), advanced age with dementia, alcoholism-related brain dysfunction (such as Wernicke encephalopathy, Korsakoff syndrome), active use of alcohol/opioids/other substance abuse, use of antibiotics/probiotics/anti-motility drugs, inability to take oral medication, unstable vital signs, transjugular intrahepatic portosystemic shunt (TIPS) or surgical shunts, liver transplantation within one-year follow-up, non-curative hepatocellular carcinoma (HCC) (≥Barcelona clinic liver cancer classification (BCLC) stage B), or active extra-hepatic malignancy were excluded.

The indications of rifaximin use in our study were persistent recurrent HE (at least two episodes within 6 months) under lactulose therapy or intolerance to lactulose due to adverse effects such as diarrhea and bloating.

Patients were enrolled for group R + L upon satisfying the above criteria and if financially feasible for the patients (cost: about USD 7600/year). All patients who received rifaximin 550 mg twice daily continuously took lactulose as a combination treatment, and lactulose dosage was adjusted to reduce intolerance. Unless severe adverse effects or economic burden occurred, there was no restriction on the use of rifaximin in our study.

On the other hand, patients receiving only lactulose (30 to 45 mL twice to four times daily) were selected as the control group (group L).

### 2.3. Data Collection

Baseline clinical parameters such as age, sex, ascites status, prior HE grade, esophageal varices (EV) status, EV bleeding (EVB) episodes, HCC, hepatorenal syndrome (HRS), spontaneous bacterial peritonitis (SBP), and comorbidities (diabetes mellitus (DM), end stage renal disease (ESRD), heart failure) were documented from medical records. The presence and severity (nil, mild, moderate, severe) of ascites were assessed by abdominal ultrasonography. Esophagogastroduodenoscopy was used for the evaluation of esophageal/gastric varices.

We also collected laboratory data, including ammonia levels, white blood cell (WBC) count, hemoglobin (Hb) level, platelet (PLT) count, prothrombin time/INR, creatinine, aspartate aminotransferase (AST), alanine aminotransferase (ALT), total bilirubin, albumin, ammonia, and model for end-stage liver disease (MELD) scores, before therapy and during follow-up.

Patients visited our outpatient clinic every 3 months, with evaluation of clinical conditions including mini-mental state examination (MMSE) and psychometric hepatic encephalopathy score (PHES) tests for cognitive function, biochemistry data, ascites condition, and any adverse events or complications.

### 2.4. Treatment Efficacy and Safety Assessments

Treatment efficacy comprises primary and secondary endpoints. The primary endpoint of our study was time to first HE recurrence (Conn score ≥ 2) within one year and was compared between the combination group (R + L) and the lactulose monotherapy group (L). The secondary endpoints included the numbers and days of hospitalization attributed to HE, the serum ammonia level during follow-up, the cognition function (MMSE), serum ammonia level, renal function, and varices condition, compared between the combination and lactulose monotherapy groups every three months up to one year. In addition, these clinical and laboratory parameters, including the cognition function MMSE score, were also compared before and after rifaximin therapy within the rifaximin add-on group.

For safety evaluation of long-term rifaximin use, we documented any discomfort, adverse events, or complications if present at follow-up.

### 2.5. Statistical Analysis

Continuous variables are expressed as median [interquartile range (IQR)]. A non-parametric Mann–Whitney U test was used to compare the continuous variables between group R + L vs. group L. Categorical variables are described as frequencies and percentages with the Chi-square test for comparison. When it came to a situation where more than 20% of data cells presented an expected frequency of <5, Fisher’s exact test was substituted for the Chi-square test. Univariable and multivariable logistic regression analysis were performed for adjusting predictors for HE recurrence within 12 months. Kaplan–Meier and Log-rank test were used for univariable survival analysis of the efficacy of group R + L vs. group L in preventing HE recurrence within 12 months in patients with cirrhosis complicated by HE. As for comparing the clinical parameters at baseline and after rifaximin add-on to lactulose initiated in group R + L patients at every visit, repeated measurement ANOVA was used. However, for clarity and simplicity, only the baseline, 6th and 12th month data comparisons were demonstrated. Statistics were performed using SPSS software (SPSS Inc., Chicago, IL, USA). A *p* value of <0.05 was considered statistically significant. Power analysis was also performed.

## 3. Results

### 3.1. Flowchart and Patient Characteristics

As shown in Figure 1, 60 patients fulfilling the inclusion criteria were included. After excluding 17 patients who met the exclusion criteria, a total of 12 patients received rifaximin add-on to lactulose combination therapy (group R + L), and 31 patients who received lactulose monotherapy (group L) were enrolled and analyzed. Demographic and baseline characteristics are shown in Table 1. Except for age, HCC status, and MELD score, there were no statistical difference between the two groups for sex, HE grade, ascites amount, EV grade, EVB, SBP, HRS, DM, ESRD, heart failure, serum ammonia, INR, WBC, Hb, PLT, creatinine, bilirubin total, AST/ALT, or albumin.

Regarding clinical parameters, the mean ages in group R + L patients were older than that in group L patients (67 vs. 58 years-old, *p* = 0.007). In group R + L, there were 4 patients diagnosed with minimal HE by PHES test, 6 with grade 1~2 HE, 2 with grade 3~4 HE. In group L, there were 21 patients diagnosed with grade 1~2 HE, while 10 patients had grade 3~4 HE by West Haven classification. There were 4 patients diagnosed with HCC in group R + L while 0 patients were diagnosed with HCC in group L. Regarding laboratory parameters, the median MELD score in group R+L was lower than that in group L (14.05 vs. 17.00, *p* = 0.048).

### 3.2. Outcomes Analysis Between Group R + L Versus Group L Primary Outcome

Upon primary endpoint analysis, the median time to first HE recurrence (Conn score ≥ 2) in the group R + L was significantly longer than that in group L during follow-up (204.50 days, IQR: 170.25–492.00 vs. 125.00 days, IQR: 42.25–247.00. *p* = 0.044). A multivariable logistic regression analysis was performed to adjust for those confounding factors. MELD score instead of Child Turcotte-Pugh (CTP) score was used in the regression model because CTP score possessed collinearity with HE grade. As shown in Table 2, independent to baseline HE grade, the odds ratio of HE recurrence within 1 year between two groups was 0.214 (*p* = 0.045), suggesting that there was a relatively lower risk of HE recurrence for patients in the group R + L compared to that in the group L. Kaplan–Meier estimates of the efficacy of group R + L vs. group L in avoiding HE recurrence within 12 months were also consistent, as shown in Figure 2.

### 3.3. Comparisons of Various Clinical and Laboratory Parameters between Group R + L vs. Group L after One-Year Follow-Up

As shown in Table 3, the median number of hospitalizations due to HE during one-year follow-up was significantly lower in group R + L vs. group L [1, (0–2) vs. 3, (2–4), *p* < 0.001]. The median days of hospitalization attributed to HE within one-year of follow-up was significantly lower in group R + L than that in group L [11, (0–27) vs. 37 (15–97), *p* = 0.003]. HE recurrence during one-year follow-up was significantly lower in group R + L than in group L. HE grade by the 12th month was also less severe in group R + L than that in group L. In addition, greater reduction in serum ammonia was noted in group R + L than that in group L after treatment for 3 months [89.0 μg/dL (59.5–111.0) vs. 169.5μg/dL (123.7–226.5), *p* = 0.002]. The effect continued to 6 months [140.3μg/dL (71.7-160.5) vs. 197.0 μg/dL (149.5–304.0), *p* = 0.007]. The reduction in ammonia did not maintain to one year.

### 3.4. Outcomes Analysis before and after Rifaximin Add-On within Group R+L

In addition, these clinical and laboratory parameters were also compared before and after rifaximin therapy within the group R + L.

For patients in group R + L, compared to baseline condition, significant improvement in clinical and laboratory parameters were noticed after rifaximin add on. As shown in Table 4 and Figure 3, the MMSE was improved both at 6 months and 12 months (*p* = 0.020). The number of hospitalizations attributed to HE also significantly decreased after rifaximin use (*p* = 0.049). The reduction in days of hospitalization attributed to HE significantly decreased at 6 months (*p* = 0.028) and lasted to 12 months (*p* = 0.046). The number of patients who suffered from EVB were also decreased significantly both at 6 and 12 months (*p* = 0.011). Moreover, serum ammonia levels also significantly decreased at 3 and 6 months (*p* = 0.0025 and *p* = 0.0085, respectively).

### 3.5. Side Effects of Rifaximin

There was no rifaximin side effects or safety complications recorded in our study.

## 4. Discussion

In this retrospective cohort study, we revealed superior treatment efficacy with rifaximin add-on to lactulose over lactulose alone in patients with cirrhosis complicated by HE at a single medical center in Taiwan. Compared with lactulose monotherapy, we found significant improvement in the maintenance of HE remission and decreased episodes and days of HE-related hospitalizations after add-on rifaximin. The serum ammonia levels were also significantly lower at the 3rd and 6th month in the add-on group. Regarding changes before and after rifaximin add-on in the same patient group, MMSE, episodes of hospitalizations, and number of patients with EVB also improved both at 6 and 12 months. The days of hospitalization and serum ammonia levels at the 3rd and 6th month were also improved. No patient discontinued rifaximin due to adverse events or complications. The above results provide a real-world evidence for the use of rifaximin add-on to lactulose instead of lactulose alone for patients with cirrhosis complicated with hepatic encephalopathy in Taiwan. Comparing the economic burden from rifaximin and the average hospital expense from one of the cirrhotic complications such as hepatic encephalopathy, variceal bleeding and ascites per patient per year in Taiwan [17,18,19], the economic burden from rifaximin is more cost-effective. Therefore, these results may justify the use and coverage of rifaximin by the National Health Insurance of Taiwan in the future.

Although the comprehensive mechanisms underlying HE are incompletely understood [2], pseudo-neurotransmitters such as gamma-aminobutyric acid (GABA) and ammonia are the best characterized neurotoxins that precipitate HE [20]. It is hypothesized that neurotoxins like ammonia, which is the product of protein metabolized by colonic bacteria and enterocyte, entered systemic circulation via portal vein and across the blood-brain barrier under cirrhotic condition, resulting in neurologic dysfunction, even HE [2,21]. The gastrointestinal tract is the primary source of ammonia while gut microbiota is increasingly recognized as another important source of ammonia. Strategies aimed to lower plasma ammonia level and improve gut dysbiosis, such as lactulose and rifaximin, are currently the mainstay of drug therapy for HE. TIPS and orthotopic liver transplantation (OLT) are recognized as the second-line treatment [20]. Other therapies under investigation include L-ornithine-L-aspartate which stimulates the metabolism of ammonia [22], polyethylene glycol (PEG) that helps the excretion of ammonia from stool by cathartic effect [23], and branched chain amino acid supplement that decreases the ratio of plasma aromatic amino acids to branched-chain amino acids [24]. In addition, probiotics containing lactobacilli and bifidobacterial etc., [25], fecal microbiota transplant (FMT) that correct microbial dysbiosis [26], are treatments that require more studies. In addition, the extracorporeal albumin dialysis (ECAD) such as molecular adsorbents recirculating system (MARS) that directly reduce endogenous toxin in case of severe HE [27,28] is served as a bridging therapy for those planned for OLT. Of note is that FMT may be the rising star in the management of HE in the future [26,29]. Since the mainstream of drug treatment for HE is lactulose and/or rifaximin [1], the latter is not routinely used in Asia including Taiwan, we therefore conducted the current study to focus on their comparison.

Preventing HE recurrence can avoid heavy burden not only to patients and their families, but also to the healthcare system and society in general [30]. HE-related hospitalization costs are still rising and have increased from $4.68 billion in 2005 to $7.25 billion in 2009 in the United States [31]. In a systematic review, rifaximin was shown to be associated with shorter hospital stays, reduced healthcare costs, and better cost-effectiveness [32]. In the current AASLD/EASL practice guideline, lactulose is recommended as a first-line pharmacologic agent for preventing recurrent HE, and rifaximin is suggested as an add-on therapy if recurrence of HE persists [1]. Kang et al. discovered that rifaximin combined with lactulose was superior to lactulose monotherapy in preventing recurrent HE in non-HCC patients [13]. However, one randomized controlled trial (RCT) found there was no difference between rifaximin and placebo in maintaining HE remission [12]. Another study found rifaximin combined with lactulose to be non-superior to lactulose monotherapy in treatment of refractory HE [14]. Obviously, the role of adding rifaximin in preventing HE recurrence is still a controversial issue. However, our study demonstrated that time to first HE recurrence was significantly longer in the rifaximin add-on group than the lactulose monotherapy group, which is a consequential finding [10]. The odds ratio, which reflects the risk of HE recurrence, was also significantly lower in patients receiving rifaximin add-on therapy. There appears to be additional benefit of rifaximin in the prevention of recurrent HE. Furthermore, previous studies have demonstrated the effect of rifaximin in reducing HE-related hospitalization [9,10,33], and our study further confirmed a decrease in number of episodes and days of hospitalization among patients receiving rifaximin and lactulose combination therapy compared to lactulose monotherapy, suggesting better cost-effectiveness (median decrease in days of hospitalizations from 37 to 11 days). In addition, there was greater reduction in serum ammonia level in the combination therapy group compared to the monotherapy group.

As we know, the development of complications from cirrhosis, such as ascites, variceal bleeding, and HE, correlates with progressively severe portal hypertension [34]. Studies have demonstrated the efficacy of rifaximin in reducing cirrhosis-related complications [13,35], and our study also reveals improvement following long-term, such as a decrease in esophageal variceal bleeding potentially linked to decreased portal hypertension.

MMSE is a widely used tool for cognitive function assessment. Although precise utility in evaluating HE is unclear, many studies employ the MMSE to evaluate cognition in cirrhotic patients with HE [36,37,38,39]. Our study also demonstrated cognitive dysfunction improvement by MMSE significantly after rifaximin add-on therapy for six months.

No patients discontinued rifaximin therapy due to adverse effects in our study. Rifaximin is safe and well-tolerated for long-term use, as is consistent with the previous reports [9,11]. Neff et al. considered rifaximin as a more cost-effective agent for HE treatment due to its efficacy on HE prevention despite its high price [40].

There are some limitations of our study. First, only a few patients were enrolled in the rifaximin add-on group, and age, HE grade, HCC status, and MELD score at baseline were not matched to the control group. However, a multivariable logistic regression analysis had been performed to adjust for these confounding factors and the result can provide evidence for conducting a larger prospective matched study in the future. Second, because of missing MMSE scores in the lactulose monotherapy group, we could not evaluate whether there was a difference in MMSE between these two groups after therapy. Third, due to retrospective design, we did not evaluate the therapeutic effect of rifaximin on the changes of sarcopenia, quality of life, or small intestine bacterial overgrowth.

In conclusion, the results of our study demonstrate that one-year efficacy of rifaximin add-on to lactulose is safe, well tolerated, and superior to lactulose alone in reducing HE recurrence and HE-related hospitalization in one-year follow-up in Taiwan. A large-scale prospective randomized control study may be warranted.

## Figures and Tables

**Figure 1 jpm-11-00478-f001:**
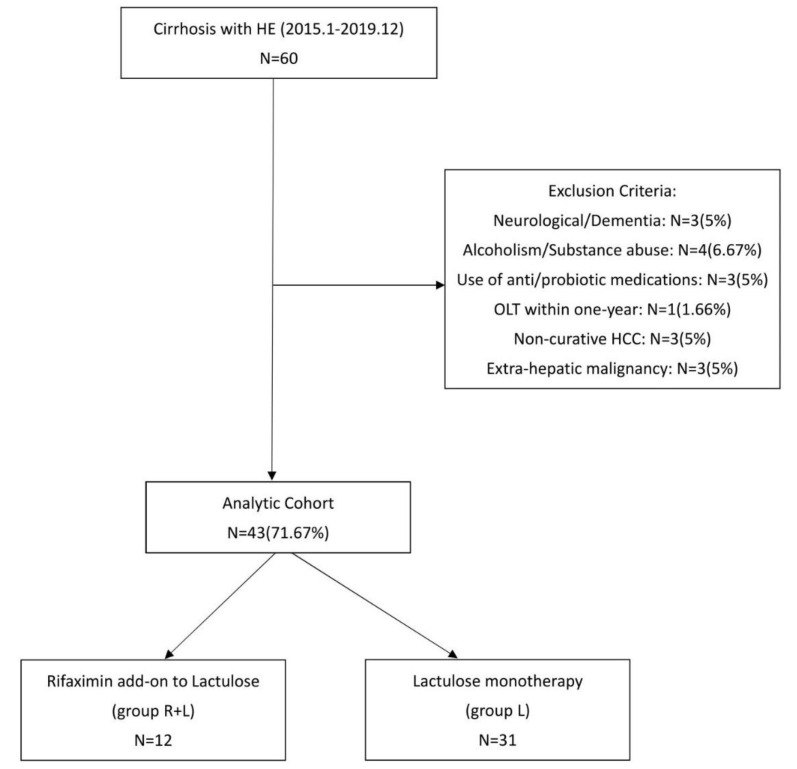
Flow chart of the study.

**Figure 2 jpm-11-00478-f002:**
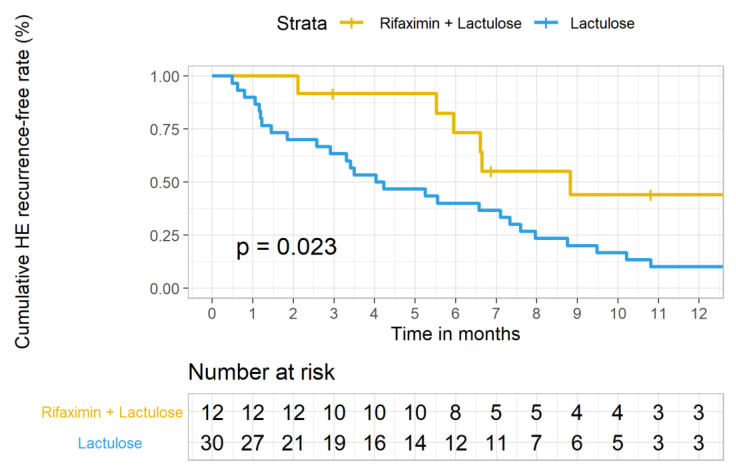
Kaplan–Meier estimates of the efficacy of rifaximin add-on to lactulose (group 1) vs. lactulose monotherapy (group 2) in preventing HE recurrence within 12 months among patients with cirrhosis complicated by hepatic encephalopathy. Time to first HE episode within 12 months of follow-up: Log-Rank *p* = 0.023.

**Figure 3 jpm-11-00478-f003:**
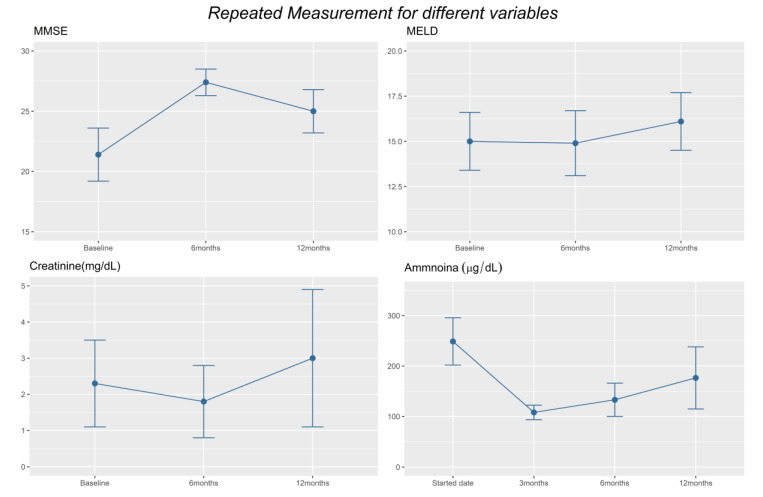
Display of important clinical and laboratory parameter changes in Group 1 patients during different follow-up time points.

**Table 1 jpm-11-00478-t001:** Demography and baseline clinical characteristics of group 1 and group 2 patients.

	Group 1 Rifaximin + Lactulose	Group 2 Lactulose	*p*-Value
Baseline Parameter	N = 12	N = 31	
**Clinical parameters**			
Age, mean ± SD	67 ± 7.95	57.58 ± 12.28	0.007
Males (%)	6 (50%)	20 (64.5%)	0.388
HE grade ^$^			0.160
Minimal/I/II/III/IV No.	4/2/4/2/0	0/6/15/9/1	
Ascites amount			0.210
Nil/Mild/Moderate/Severe No.	7/2/2/1	13/4/6/8	
EV			0.052
Nil/F1/F2/F3 No.	1/3/3/5	10/8/8/5	
EVB No. (%)	6 (50%)	7 (22.6%)	0.108
SBP No. (%)	0 (0%)	0 (0%)	
HRS No. (%)	0 (0%)	0 (0%)	
HCC No. (%)	4 (33.3%)	0 (0%)	<0.001
DM No. (%)	5 (41.7%)	7 (22.5%)	0.216
ESRD No. (%)	1(8.3%)	5 (16.1%)	0.513
Heart Failure No. (%)	1 (8.3%)	1 (3.2%)	0.481
**Laboratory parameters** **Median (IQR)**			
Ammonia (μg/dL)	226.0 (186.40–262.0)	224.0 (174.75–317.0)	0.895
MELD	14.05 (11.87–16.55)	17.0 (14.0–22.0)	0.048
INR	1.40 (1.30–1.60)	1.50 (1.30–1.70)	0.874
WBC (10^3^/uL)	3.45 (2.75–5.37)	4.85 (3.50–6.35)	0.060
Hemoglobin (g/dL)	9.55 (8.82–10.30)	10.0 (8.65–12.17)	0.549
PLT (10^3^/uL)	62.0 (37.0–79.75)	81.0 (59.0–111.0)	0.091
Creatinine (mg/dL)	0.95 (0.83–1.09)	1.13 (0.63–1.98)	0.924
Bilirubin, Total (mg/dL)	1.85 (1.30–2.12)	2.50 (1.30–3.30)	0.095
AST (U/L)	40.0 (34.0–53.0)	51.0 (37.0–73.0)	0.188
ALT (U/L)	22.0 (20.0–25.0)	25.50 (18.75–31.50)	0.461
Albumin (g/dL)	3.0 (2.55–3.38)	2.95 (2.40–3.36)	0.987

^$^ West-Heaven classification [Conn H.O., Liberathal M.M. The hepatic coma syndromes and lactulose. Williams and Wilkins 1979; 1–121]. [DOI:10.1016/0016-5085(79)90191-4]. EV: esophageal varices; F1: form 1; EVB: esophageal variceal bleeding; SBP: spontaneous bacterial peritonitis; HRS: hepatorenal syndrome; HCC: hepatocellular carcinoma; DM: diabetes mellitus; ESRD: end-stage-renal disease. MELD: model for end-stage liver disease; INR: international normalized ratio; WBC: white blood cells; PLT: platelet; AST: aspartate transaminase; ALT: alanine transaminase.

**Table 2 jpm-11-00478-t002:** Logistic regression estimates of baseline parameters for predicting HE recurrence within 1 year.

	Univariable Logistic Reg.		Multivariable Logistic reg.	
Variables	OR (95% CI)	*p*-Value	OR (95% CI)	*p*-Value
Group (rifaximin + lactulose vs. lactulose)	0.148 (0.032, 0.694)	0.015	0.214 (0.037, 0.925)	0.045
Age	0.985 (0.927, 1.047)	0.628		
Sex	1.719 (0.377, 7.849)	0.484		
MELD	1.160 (0.969, 1.389)	0.105		
HE grade (≥II vs. I and minimal)	13.067 (2.5000, 68.291)	0.002	10.182 (1.793, 57.802)	0.009
EVB (n, %)	6 (50%)	0.832		
HCC	3.875 (0.471,31.912)	0.208		
DM	2.083 (0.467, 9.288)	0.336		
ESRD	1.812 (0.280, 11.750)	0.533		
Heart Failure	3.556 (0.202, 62.632)	0.386		

MELD: model for end-stage liver disease; HE: hepatic encephalopathy; EVB: esophageal variceal bleeding; HCC: hepatocellular carcinoma; DM: diabetes mellitus; ESRD: end-stage-renal disease.

**Table 3 jpm-11-00478-t003:** Comparisons of clinical outcomes and laboratory parameters between groups after one-year follow-up.

Clinical Parameters	Group 1 Rifaximin + Lactulose	Group 2 Lactulose	*p*-Value
Number of hospitalizations due to HE	1 (0.0, 2.0)	3 (2.0, 4.0)	<0.001
Days of hospitalizations	11 (0.0, 27.0)	37 (15.0, 97.0)	0.003
HE Recurrence	6 (50%)	27 (87.1%)	0.011
HE grade ^$^ (n)	7	23	0.003
Minimal/I/II/III/IV/V:	1/0/0/1/0/5	9/2/6/4/2/0	
Death (n, %)	2 (20%)	13 (41.9%)	0.216
MMSE	25.0 ± 4.1		
EVB (n, %)	6 (50%)	31 (100%)	0.313
Ascites (n)	7	21	0.307
Nil/Mild/moderate/Severe:	5/1/0/1	10/5/1/5	
**Laboratory parameters**			
Serum Ammonia	132.2 (62.0, 222.0)	201 (157.0, 239.0)	0.179
MELD	14.6 (12.0, 20.1)	20 (15.0, 22.0)	0.115
INR	1.4 (1.4, 1.5)	1.5 (1.3, 1.9)	0.616
WBC (10^3^/uL)	4.1 (3.6, 6.2)	4.3 (2.9, 8.1)	0.772
Hemoglobin (g/dL)	65 (47.0, 94.0)	78 (49.0, 127.0)	0.825
PLT (10^3^/uL)	9.5 (8.9, 11.5)	9.85 (8.5, 10.8)	0.386
Creatinine (mg/dL)	1.165 (0.8, 1.8)	1.39 (0.9, 3.4)	0.662
Bilirubin Total (mg/dL)	2.6 (1.0, 3.0)	1.9 (1.0, 4.5)	0.934
AST (U/L)	43 (29.0, 47.0)	48 (36.0, 55.0)	0.137
ALT (U/L)	26 (19.0, 46.0)	26 (21.0, 37.0)	1.0
Albumin (g/dL)	2.805 (2.6, 3.0)	2.94 (2.6, 3.4)	0.861

^$^ West-Heaven classification [Conn H.O., Liberathal M.M. The hepatic coma syndromes and lactulose. Williams and Wilkins 1979; 1–121 [DOI:10.1016/0016-5085(79)90191-4].

**Table 4 jpm-11-00478-t004:** Comparisons of clinical and laboratory parameters at baseline and after rifaximin plus lactulose initiated in patients (N = 12) during different follow-up visits.

	Baseline	6 Months	12 Months	*p*-Value
**Clinical parameters**				
MMSE	21.4 ± 2.2	27.4 ± 1.1	25.0 ± 1.8	0.02
EVB	50%	0%	0%	0.011
Ascites	41.70%	36.40%	28.60%	0.794
**Lab parameters**				
MELD	15.0 ± 1.6	14.9 ± 1.8	16.1 ± 1.6	0.524
INR	1.4 ± 0.03	1.4 ± 0.03	1.4 ± 0.05	0.484
WBC	4.0 ± 0.5	4.7 ± 0.8	4.8 ± 0.5	0.352
Hb	9.8 ± 0.4	10.0 ± 0.5	9.9 ± 0.5	0.909
Plt	71.1 ± 10.8	79.0 ± 9.9	74.1 ± 12.4	0.566
Cr	2.3 ± 1.2	1.8 ± 1.0	3.0 ± 1.9	0.273
Bilt	1.8 ± 0.4	2.2 ± 0.4	2.2 ± 0.5	0.268
Alb	3.0 ± 0.2	3.0 ± 0.2	2.9 ± 0.1	0.45
Serum Ammonia	248.9 ± 47.0	133.1 ± 33.0	176.5 ± 61.6	0.187 ^†^

† *p*-value of serum ammonia by repeated measure ANOVA uses 4 time-points including at 3 months with mean and SD of 108.0 ± 14.4. Two patients took rifaximin for less than 12 months, and one for less than 6 months.

## Data Availability

Data will be available under readers request if indicated.

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
