# Peer review of "Real-World Experience of the One-Year Efficacy of Rifaximin Add-On to Lactulose Is Superior to Lactulose Alone in Patients with Cirrhosis Complicated with Recurrent Hepatic Encephalopathy in Taiwan"

_jpm, 2021, doi:10.3390/jpm11060478_

Round 1

Reviewer 1 Report

Chang C. et al. wrote about an interesting subject, concerning chronic hepatic encephalopathy and treatment. They focused on association between Rifaximin and Lactulose.

The main limitation is reported by the authors and concerns the weak number of subjects. It impacts on differences between the two groups. Furthermore, interpretation of results is quite limited.

By the way, there are some revisions point:

Please, translate in US dollars (or Euros.) Taiwan money.

Impact on economic burden would be interesting, such as quality of life index.

Author Response

Point 1: Please, translate in US dollars (or Euros.) from Taiwan money.

Response 1: Thanks for the valuable comments. The cost of rifaximin takes NTD 216,000/year which translates to USD 7,600/year according to current exchange rate. We also revised it with mark in the text.

Point 2: Impact on economic burden would be interesting, such as quality of life index

Response 2: Thanks for the valuable comments. Comparing the economic burden from rifaximin and the average hospital expense from one of the cirrhotic complications such as hepatic encephalopathy, variceal bleeding and ascites per patient per year in Taiwan (Hsieh, Chee-Ruey, et al. Journal of Clinical Gastroenterology 38.10 (2004): S148-S152.) (Lin, W‐A., et al. Alimentary Pharmacology & Therapeutics 24.10 (2006): 1483-1493.) (El Khoury, Antoine C., et al. Journal of Medical Economics 15.5 (2012): 887-896.), the economic burden from rifaximin is more cost-effective. The use of rifaximin could reduce the recurrence of HE and other cirrhotic complications in Taiwan as we demonstrated in this study. Therefore, one of our research goals is to justify the use and coverage of rifaximin by the National Health Insurance of Taiwan. However, due to the retrospective cohort design, we did not perform evaluation such as quality of life index.

Reference

  1. Hsieh, Chee-Ruey, and Chuei-Wen Kuo. "Cost of chronic hepatitis B virus infection in Taiwan." Journal of Clinical Gastroenterology 38.10 (2004): S148-S152.
  2. LIN, W‐A., Y‐H. TARN, and S‐L. TANG. "Cost–utility analysis of different peg‐interferon alpha‐2b plus ribavirin treatment strategies as initial therapy for naïve Chinese patients with chronic hepatitis C." Alimentary Pharmacology & Therapeutics 24.10 (2006): 1483-1493.
  3. El Khoury, Antoine C., et al. "Economic burden of hepatitis C-associated diseases: Europe, Asia Pacific, and the Americas." Journal of Medical Economics 15.5 (2012): 887-896.

Reviewer 2 Report

The presented manuscript Targets the Question if long-term rifaximin+lactulose is superior to Lactulose alone. Unfortunately concepualization and performance of the study does not allow to answer the question. There are many major points to be addressed:

  1. The question is not quite new and there are some studies covering this questions and many centers world wide already perform long-term combination therapy. Therefore, it is mainly a local question.
  2.  The authors correctly describe that quidelines recommend a 3-6 month combination therapy and the raised question is, if a Long-term therapy is superior. To answer this question all patients should have had the combination for 6 month and should then be randomized into laculose vs. combination for at least another 6-12 month to really address this question. In the presented study 37$ of the patients were not treated according to the quidelines but were on Lactulose monotherapy from the beginning on!
  3.  The authors did not provide sufficient Information how the patients were stratified to which therapy arm. Was it random?
  4.  The numbers of patients are way o small to answer the Question. There were only 12 patients in the combination therapy arm and 31 in the control Group.
  5.  Why only 1/4 of the patients was "randomized" into the combination therapy arm. 
  6.  Why patients with TIPS were excluded? Probably the study numer was to small for matching of these patients?
  7.  As a minor point the style in brackets (BCLC..) is different from the rest of the text style

Author Response

Point 1: The question is not quite new and there are some studies covering this questions and many centers worldwide already perform long-term combination therapy. Therefore, it is mainly a local question.

Response 1: Thanks for the good comments. Although there are already many studies demonstrating the efficacy of long-term combination therapy, but the duration of the combination treatment from most of them were 6 months (Hudson, Mark, et al. European Journal of Gastroenterology & Hepatology 31.4 (2019): 434.), and the guideline from AASLD/EASL is not revised since 2014. Therefore, we demonstrated the extended efficacy of combination therapy up to one year in Taiwan. As you just mentioned, the evidence to support the combination therapy in Taiwan is scarce, and in the study we have provided some evidence to support the use and coverage of rifaximin by the National Health Insurance of Taiwan, as Taiwan is well known for its health care system (Wu, Tai-Yin, et al. London Journal of Primary Care 3.2 (2010): 115-119.) and a high prevalence rate of viral hepatitis like HBV, HCV (Lu, Sheng‐Nan, et al. International Journal of Cancer 119.8 (2006): 1946-1952.) and its related complications in Taiwan.

Point 2: The authors correctly describe that guidelines recommend a 3-6 month combination therapy and the raised question is, if a Long-term therapy is superior. To answer this question all patients should have had the combination for 6 month and should then be randomized into lactulose vs. combination for at least another 6-12 month to really address this question. In the presented study 37% of the patients were not treated according to the guidelines but were on lactulose monotherapy from the beginning on!

Response 2: Thanks for the valuable comments. Almost all patients in Taiwan take medicine paid by the national health insurance(https://www.taiwannews.com.tw/en/news/4106372). Although the guideline recommends rifaximin as an add-on to lactulose for prevention of recurrent episodes of HE after the second episode, it is currently not supported by the national health insurance administration in Taiwan. We provided alternative treatments for rifaximin such as neomycin since rifaximin is less available. Therefore, one of our research goals is to justify and support the use and coverage of rifaximin by the National Health Insurance of Taiwan in the future.

Point 3: The authors did not provide sufficient Information how the patients were stratified to which therapy arm. Was it random?

Response 3: Thanks for the good comments. It was a retrospective cohort study and we had mentioned it in the “method section of the abstract”, in the “study design section in the materials and methods”, and the “limitations section in the discussion”. Although it was not randomly assigned, there were no statistical difference between the two groups for baseline data such as sex, HE grade, ascites amount, EV grade, EVB, SBP, HRS, DM, ESRD, heart failure, serum ammonia, INR, WBC, Hb, PLT, creatinine, bilirubin total, AST/ALT, or albumin, except for age, HCC status, and MELD score.

Point 4: The numbers of patients are way to small to answer the Question. There were only 12 patients in the combination therapy arm and 31 in the control Group.

Response 4: Thanks for the good comments. We have performed power analysis to get the power. Although the power is not high enough (nearly 0.7), there is still some evidence to prove combination therapy is superior to monotherapy prevention of HE recurrence.

We hope the results could help to justify and support the use and coverage of rifaximin by the National Health Insurance of Taiwan in the future.

Point 5: Why only 1/4 of the patients was "randomized" into the combination therapy arm.

Response 5: Thanks for the valuable comments. It was a retrospective cohort study and we had mentioned it in the “method section of the abstract”, in the “study design section in the materials and methods”, and the “limitations section in the discussion”

Only 1/4 of the patients received combination therapy because almost all patients in Taiwan take medicine paid by the national health insurance (https://www.taiwannews.com.tw/en/news/4106372), but the rifaximin is not supported by the national health insurance administration in Taiwan.  Although it was not randomly assigned, there were no statistical difference between the two groups for baseline data such as sex, HE grade, ascites amount, EV grade, EVB, SBP, HRS, DM, ESRD, heart failure, serum ammonia, INR, WBC, Hb, PLT, creatinine, bilirubin total, AST/ALT, or albumin, except for age, HCC status, and MELD score.

Point 6: Why patients with TIPS were excluded? Probably the study number was to small for matching of these patients?

Response 6: Thanks for the good comments. TIPS is seldom performed in Taiwan (Lo, Gin-Ho. Journal of the Chinese Medical Association 77.8 (2014): 395-402.) due to the easy access of variceal ligation or ascites tapping. In addition, we want to reduce confounding factors in our study such as TIPS -related HE.

Point 7: As a minor point the style in brackets (BCLC.) is different from the rest of the text style

Response 7: Thanks for the good comment. We have revised it with mark in the text.

Reference

  1. Hudson, Mark, and Marcus Schuchmann. "Long-term management of hepatic encephalopathy with lactulose and/or rifaximin: a review of the evidence." European Journal of Gastroenterology & Hepatology 31.4 (2019): 434..
  2. Wu, Tai-Yin, Azeem Majeed, and Ken N. Kuo. "An overview of the healthcare system in Taiwan." London Journal of Primary Care 3.2 (2010): 115-119.
  3. Lu, Sheng‐Nan, et al. "Secular trends and geographic variations of hepatitis B virus and hepatitis C virus‐associated hepatocellular carcinoma in Taiwan." International Journal of Cancer 119.8 (2006): 1946-1952.
  4. Lo, Gin-Ho. "The use of transjugular intrahepatic portosystemic stent shunt (TIPS) in the management of portal hypertensive bleeding." Journal of the Chinese Medical Association 77.8 (2014): 395-402.

Round 2

Reviewer 1 Report

Authors answered accurately to the comments. I have no other suggestion to add.

Author Response

Point 1: Authors answered accurately to the comments. I have no other suggestion to add.

Response 1: Thanks for the valuable comments.

Reviewer 2 Report

The raised points were discussed in the point-to-point reply. However, the manuscript has not been altered. 

Author Response

Point 1: The raised points were discussed in the point-to-point reply. However, the manuscript has not been altered.

Response 1: Thanks for the valuable comments. We have added all the changes underlined in the manuscript as you recommended.
